# Pan-mammalian analysis of molecular constraints underlying extended lifespan

**Amanda Kowalczyk[1,2]\*, Raghavendran Partha[1,2], Nathan L Clark[2,3,4], Maria Chikina[2]\***

[1]Joint Carnegie Mellon University-University of Pittsburgh PhD Program in Computational Biology, Pittsburgh, United States; [2]Department of Computational and Systems Biology, University of Pittsburgh, Pittsburgh, United States; [3]Pittsburgh Center for Evolutionary Biology and Medicine, University of Pittsburgh, Pittsburgh, United States; [4]Department of Human Genetics, University of Utah, Salt Lake City, United States

**Abstract** Although lifespan in mammals varies over 100-fold, the precise evolutionary mechanisms underlying variation in longevity remain unknown. Species-specific genetic changes have been observed in long-lived species including the naked mole-rat, bats, and the bowhead whale, but these adaptations do not generalize to other mammals. We present a novel method to identify associations between rates of protein evolution and continuous phenotypes across the entire mammalian phylogeny. Unlike previous analyses that focused on individual species, we treat absolute and relative longevity as quantitative traits and demonstrate that these lifespan traits affect the evolutionary constraint on hundreds of genes. Specifically, we find that genes related to cell cycle, DNA repair, cell death, the IGF1 pathway, and immunity are under increased evolutionary constraint in large and long-lived mammals. For mammals exceptionally long-lived for their body size, we find increased constraint in inflammation, DNA repair, and NFKB-related pathways. Strikingly, these pathways have considerable overlap with those that have been previously reported to have potentially adaptive changes in single-species studies, and thus would be expected to show decreased constraint in our analysis. This unexpected finding of increased constraint in many longevity-associated pathways underscores the power of our quantitative approach to detect patterns that generalize across the mammalian phylogeny.

**\*For correspondence:**
kowaae22@pitt.edu (AK);
mchikina@pitt.edu (MC)

**Competing interests:** The authors declare that no competing interests exist.

## Introduction

Humans age in the sense that an individual's probability of dying increases as a function of time lived. Interestingly, this trend is not true of all species but is true of mammals generally (*Jones et al., 2014*). Numerous hypotheses of mammal-specific aging exist, including the antagonistic pleiotropy hypothesis (*Williams, 1957*) and the mutation accumulation hypothesis (*Medawar, 1952*), both of which refer to changes within an individual throughout its lifetime that result in aging. The antagonistic pleiotropy hypothesis postulates that genes that are beneficial to early life become detrimental later in life. Such genes are retained because they increase early-life reproductive output and thus increase fitness. The mutation accumulation hypothesis predicts that a gradual accumulation of errors in DNA sequence as a result of repeated replication during a lifetime's worth of cell divisions will lead to a gradual breakdown of functionality. Support for both hypotheses has been found in individual species. Recent work has identified disease-related SNPs in age-related genes that are beneficial in early life and detrimental in later life in humans, thus indicating selective pressures associated with gene evolution related to aging and supporting the antagonistic pleiotropy hypothesis (*Rodríguez et al., 2017*). A study of SNVs in the human brain found that number of mutations was positively correlated with age and that mutations were at loci associated with age-related disease,

thus supporting the mutation accumulation hypothesis (*Lodato et al., 2018*). However, in human populations, variability in the aging phenotype is limited and many confounding biological changes are correlated with aging, making it difficult to pinpoint specific biological processes that are causal and thus amenable to manipulation.

On the other hand, lifespan varies dramatically (>100 fold) across mammals (*Nowak, 1999*), making comparative genomics a fruitful avenue for aging research. Numerous studies have investigated the genomic features of mammals with extreme lifespan such as bats (*Foley et al., 2018*; *Seim et al., 2013*), naked mole-rats (*Kim et al., 2011*), whales (*Keane et al., 2015*), and elephants (*Sulak et al., 2016*) to identify potential causative genetic changes. In *Myotis*, the longest-lived bat genus, species show lack of telomere shortening and corresponding expression changes in telomere maintenance and DNA repair genes (*Foley et al., 2018*). Comparative genomics studies have also suggested that changes in the insulin growth factor one pathway may enable increased lifespan and cancer resistance in *Myotis brandtii* (Brandt's bat) (*Seim et al., 2013*). Similarly, naked mole-rats show differential regulation of genes associated with macromolecule degradation, mitochondrial function, and TERT, a gene associated with telomere maintenance, as well as changes to genes related to tumor suppression (*Kim et al., 2011*). In primates, genes associated with cardiovascular function, coagulation, and healing have been demonstrated to show evolutionary correlations with lifespan (*Muntané et al., 2018*). Sequencing of the bowhead whale genome revealed species-specific changes in DNA repair, cell cycle, and aging genes (*Keane et al., 2015*). In elephants, a striking increase in TP53 copy number has been linked to increased cancer resistance enabling longer lifespan (*Sulak et al., 2016*). Despite compelling results, these single- and limited-species studies have limitations. The species studied differ from their nearest sequenced relatives in multiple physiological traits as well as millions of nucleotides. Thus, while single-species studies have yielded some credible candidates for genes associated with increased lifespan, it is difficult to know to what extent these represent insights into the universal mechanisms of lifespan regulation rather than species-specific adaptation or coincidental neutral changes. In this study, we develop new methodology to evaluate the relationship between the evolutionary constraint of genes and pathways and quantitative lifespan traits in an unbiased, genome-wide, pan-mammalian analysis.

The wide range of lifespans across the mammalian phylogeny (*Figure 1A*) provides the ideal dataset to investigate lifespan from a comparative genomics perspective. Because independent changes in lifespan occurred repeatedly in the mammalian species tree, lifespan can be viewed as a convergent trait. Molecular features that correlate with convergent changes in lifespan therefore may also occur repeatedly across a variety of organisms. In our study we use protein evolutionary rates quantified as the number of amino acid substitutions on a phylogenetic branch to infer convergent rate shifts associated with lifespan traits across the mammalian phylogeny.

Evolutionary rates are useful for linking phenotypes to genes because they reflect evolutionary constraint experienced by a genetic element (*Zhang and Yang, 2015*). In the absence of diversifying selection, genetic elements that support a specific trait are expected to be more constrained in species where the trait has a larger contribution to fitness. In agreement with this expectation, multiple studies have shown that a genetic element providing a function less important for a given species is under less constraint and hence exhibits a faster evolutionary rate (*Clark et al., 2013*; *Janiak et al., 2018*; *Roscito et al., 2018*; *Wertheim et al., 2015*). Reciprocally, when an element becomes relatively more important, its rate is expected to slow. Thus, in cases of phenotypic convergence, rates can be exploited to reveal important genes associated with the phenotype, such as changes to muscle and skin genes associated with the mammalian transition to a marine environment (*Chikina et al., 2016*) and loss of constraint of vision-related genetic elements in subterranean mammals (*Partha et al., 2017*; *Prudent et al., 2016*). Rate shifts can thus provide an evolutionary perspective on the contribution of genes, non-coding elements, or pathways to phenotypes of interest (*Hiller et al., 2012*). Here we report the genome-wide, pan-mammalian correlations between evolutionary rates of genes and lifespan phenotypes.

## Results

In mammals, lifespan is strongly positively correlated with adult body size such that the largest mammals (whales) are longest-lived and the smallest mammals (small rodents) are shortest lived (*Figure 1B*). However, if lifespan is corrected for body size, species including bats, the naked mole-

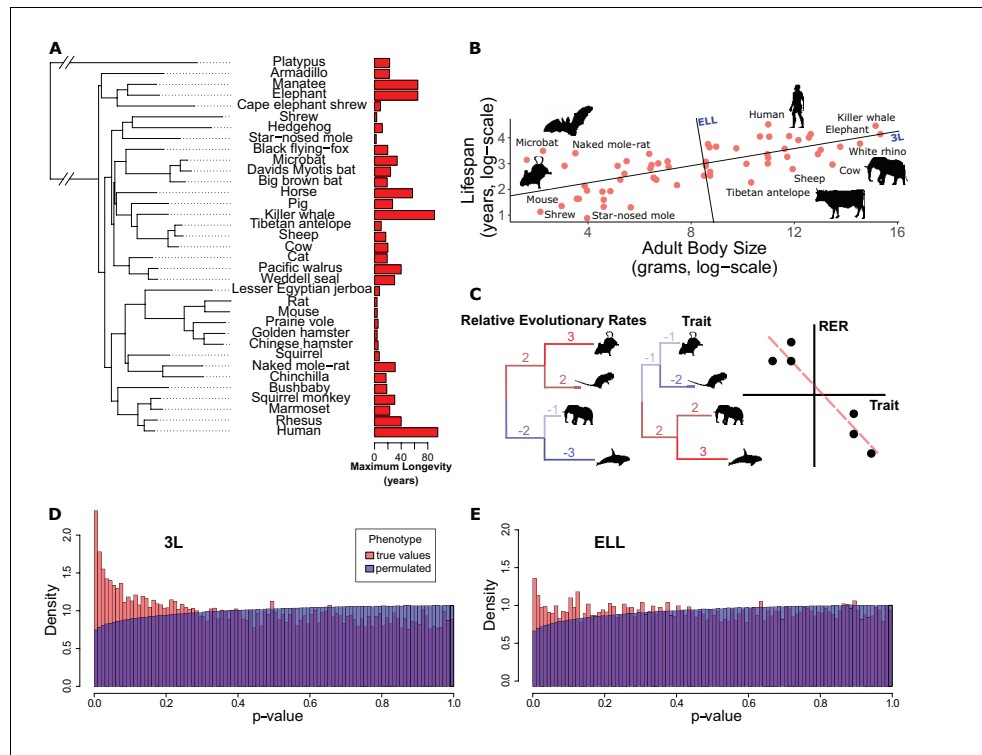

**Figure 1.** Many genes have evolutionary rates correlated with longevity phenotypes as demonstrated by analysis with RERconverge. (A) A subset of species used for this analysis alongside their maximum longevity values. Lifespan varies widely across mammals independent of phylogeny. (B) Mammal body size and maximum lifespan values for 61 species. Lines represent the 3L phenotype and the ELL phenotype (also see *Figure 1—figure supplement 1*). (C) RERconverge pipeline to find correlation between relative evolutionary rates of genes and change in lifespan phenotypes. (D and E) Distribution of p-values from correlations between evolutionary rates of genes and change in the 3L and ELL phenotypes indicate an enrichment of significant correlations (also see *Figure 1—figure supplement 11*).

The online version of this article includes the following figure supplement(s) for figure 1:

**Figure supplement 1.** 3L phenotype values (A) and ELL phenotype values (B) for 61 mammal species alongside mammalian phylogenetic tree.

**Figure supplement 2.** Correlation statistics for genes and enrichment statistics for canonical pathways plotted with statistics calculated from data with bat and naked mole-rat removed (A, B, C, and D) and with marine species removed (E, F, G, and H).

**Figure supplement 3.** Each panel demonstrates the correlation between results using all species and results with ten to eighty percent of species removed.

**Figure supplement 4.** Scatterplots for both 3L and ELL phenotype trait change versus relative evolutionary rate.

**Figure supplement 5.** Diagram of a toy example of permulation calculations.

**Figure supplement 6.** Quantile-quantile plots demonstrating that permulation p-values are more conservative than permutation p-values for both 3L and ELL phenotypes (A and B) and permulation p-values are equally as conservative as simulation p-values for both 3L and ELL phenotypes (C and D).

**Figure supplement 7.** Phylogenetic tree with all 61 mammal species used for RERconverge analysis.

**Figure supplement 8.** Phylogenetic tree with 34 placental mammal species used for branch-site tests for positive selection.

**Figure supplement 9.** Alternative tree topologies used to test for robustness to phylogeny topology errors and incomplete lineage sorting.

**Figure supplement 10.** Correlations between gene correlation and pathway enrichment statistics between alternative tree topologies and the Meredith+ tree topology used for all other analyses.

**Figure supplement 11.** Q-Q plots demonstrating the relationship between null gene permulation p-values and a standard uniform distribution and theoretical gene p-values and a standard uniform distribution for both 3L and ELL phenotypes.

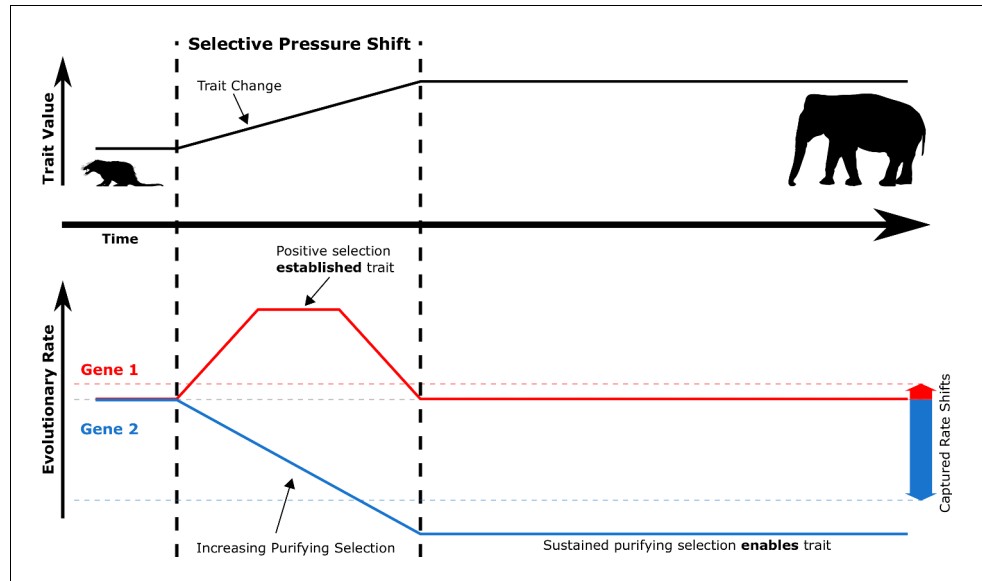

**Figure 2.** During selective pressure shifts that drive phenotypic changes, the genetic evolutionary rate landscape shifts for many genes. Imagine a simplified case where Gene 1 and Gene 2 start at the same evolutionary rate (solid red and blue lines, respectively). Trait-driving genes like Gene 1 enter a transient state of increased evolutionary rates to drive trait change. Complementary genes that support the trait like Gene 2 experience increased purifying selection to allow the trait to persist as it is established. True rates (solid red and blue lines) are not measurable because they represent rates at inaccessible ancestral evolutionary timepoints – only accessible extant sequences can be used to access average rates over time (dashed red and blue lines, which represent positive and negative RERs). Therefore, transient periods of positive selection are less readily able to be captured by RERconverge than sustained purifying selection due to the greater magnitude of their captured rate shifts.

rat, and some primates are clearly exceptionally long-lived given their body sizes. The term 'longevity' is applied to both phenotypes and previous studies have focused on small numbers of species at both phenotypic extremes to find genetic and physiological explanations for the extended phenotypes (*Foley et al., 2018*; *Keane et al., 2015*; *Kim et al., 2011*; *Seim et al., 2013*; *Sulak et al., 2016*). In our study we explicitly distinguish two different extended longevity traits: the 'long-lived large-bodied' trait (3L) and the 'exceptionally long-lived given body size' trait (ELL). Using maximum lifespan and body size data (*Tacutu et al., 2018*) we define the 3L and ELL phenotypes (*Supplementary file 1*) to be the first and second principal components of body size and maximum lifespan (*Figure 1B*, *Figure 1—figure supplement 1*). The resulting trait values are orthogonal with respect to each other, ensuring that they can be analyzed independently.

Having defined the 3L and ELL phenotypes, we compute the association between these phenotypes and protein-specific relative evolutionary rates (RERs) using the RERconverge method (*Kowalczyk et al., 2018*; *Partha et al., 2018*) (*Figure 1C* and Materials and methods). Relative evolutionary rates quantify the deviation in evolutionary rate of a protein along a specific phylogenetic branch from proteome-wide expectations. Negative RERs indicate fewer substitutions than expected due to increased constraint. Positive RERs correspond to more substitutions than expected, which could arise due to relaxation of constraint or positive selection.

After computing correlations between all protein relative evolutionary rates and the 3L and ELL phenotypes, we find an excess of low p-values (*Figure 1D* and *Figure 1E*). In order to evaluate how much of the signal is due to true association with the phenotypes, we used a phylogenetically restricted permutation strategy (termed 'permulations', see Materials and methods) to generate null correlation statistics. We find that the real and permuted p-value distributions are indeed different, indicating that a large fraction of genes are correlated with lifespan phenotypes. We quantify the fraction of non-null genes using the $\pi_1$ method (*Storey, 2003*). Using our permulation p-values as the null distribution, the fraction of true positives was inferred to be ~15% ($\pi_1 = 0.153$) for the 3L phenotype and ~7.5% ($\pi_1 = 0.075$) for the ELL phenotype. Using the theoretical uniform null

distribution, the corresponding values are $\pi_1=0.108$ and $\pi_1 = 0.021$, respectively. Regardless of the null distribution choice, our analysis clearly demonstrates a significant molecular signal for gene evolutionary rates correlated with lifespan phenotypes, with a considerably higher number of associations for the 3L phenotype.

Our analysis investigates both positive and negative correlations between evolutionary rates of genes and changes in lifespan phenotypes. Positive correlations represent genes with faster evolutionary rates in species with high 3L and ELL phenotype values relative to species with low phenotype values. Conversely, negative correlations represent genes with slower evolutionary rates in species with high 3L and ELL phenotype values relative to species with low phenotype values. Note that correlation directionality is relative – faster evolution in low phenotype values corresponds to slower evolution in high phenotype values and vice versa. However, the choice has important consequences for biological interpretation and necessitates an in-depth discussion.

RERs reflect the amount of constraint on the genetic element. A decrease in RER implies greater evolutionary constraint and a greater contribution of that gene to fitness. An increase in RER, on the other hand, has two nearly opposite interpretations. An increased RER could arise because of a relaxation of evolutionary constraint driven by a reduced contribution to fitness. Alternatively, an increased RER could arise due to positive (also termed directional) selection, which implies that that the gene is actively undergoing directed evolution and thus could be contributing to trait-related innovation. However, since there is no default evolutionary rate for protein coding sequence, genes evolving *slower* in *long-lived* species could just as easily be interpreted as evolving *faster* in *short-lived* species. In our study, we have chosen to interpret the rate changes with reference to the effect in long-lived species (*Figure 2*). The ancestral mammal is believed to have been small and short-lived, and thus large values of 3L and ELL are derived traits. Consequently, our interpretation reduces to assuming that change in phenotype and shifts in evolutionary rate coincide.

With this interpretation, negative correlations imply that a gene is more important in species with large values of 3L and ELL, while positive correlations imply either a relaxation of constraint or positive selection in species with large values of 3L and ELL. While such positively correlated genes would be of great interest if they indeed represent molecular innovations underlying evolution of extended lifespan, we find relatively fewer such genes, no evidence of positive selection among them (see *Supplementary file 6*), and no enriched pathways associated with them.

We thus focus our analysis on the negatively correlated genes, which we interpret as being under increased purifying selection in species with high longevity (3L or ELL) values. It is theoretically possible for negative correlations to be caused by accelerated evolution in small and short-lived species because of directional selection associated with development of low longevity values. However, this possibility can be rules out via branch-site models for positive selection using the low longevity species as foreground, as these show little evidence for positive selection (see *Supplementary file 6*). Together, these analyses support a single interpretation of the main rate convergence signal as a *decrease* in evolutionary rate, and thus an *increase* in purifying selection, experienced by species with large 3L and ELL values. It is important to emphasize that these genes are unlikely to have contributed to molecular innovation that lead to the establishment of 3L and ELL traits, but rather these represent existing biological systems that become especially important after the traits are established (see Discussion).

While we observe a clear excess of genes at low p-values, we focus on pathway enrichment analysis which both demonstrates a stronger signal and facilitates interpreting our results in the context of existing knowledge. We investigate enriched pathways for both 3L and ELL phenotypes using a rank-based method (*Supplementary file 2*). After performing standard multiple-hypothesis testing corrections on the empirical p-values from permulations, there remains considerable pathway-level signal underlying the 3L and ELL traits. Both the gene-level and pathway-level results were highly robust to species removal, which indicates the biological pathways revealed here are important for longevity across mammals and are not restricted to specific species (see Materials and methods, *Figure 1—figure supplement 2*, *Supplementary file 4*, and *Supplementary file 5*).

Among pathways under increased constraint in 3L species, we find a striking abundance of pathways related to cancer control. Those pathways can be organized into the broad categories of 'cell cycle control', 'cell death', and 'innate and adaptive immunity', and they also include other cancer-related pathways such as p53 regulation and telomere maintenance (*Figure 3A*). We likewise see a significant enrichment in cancer-related genes more broadly. We compared correlation statistics

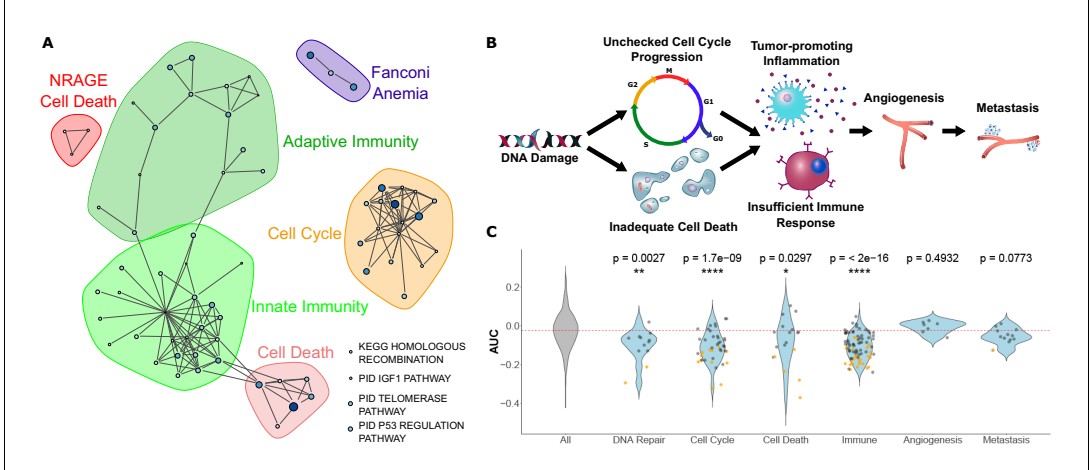

**Figure 3.** Pathways that evolve slower in long-lived, large-bodied mammals are related to control of cancer. (A) Significantly enriched pathways under increased constraint in species with larger values of the 3L phenotype. Each dot represents a pathway, and the size and color of the dot represents the negative log of the rank-sum enrichment statistic. Width of lines connecting pathways represent the number of genes the pathways have in common. (B and C) Pathways under increased constraint in 3L species play various roles in cancer control. Pathways associated with early stages of cancer development (DNA repair, cell cycle control, cell death, and immune functions) are significantly enriched, while pathways for later stages of cancer development (angiogenesis and metastasis) are not enriched. In C), each dot represents a pathway. Yellow dots have significant permulation p-values while black dots do not. Note that dots for 'All' pathways are excluded for the sake of clarity.

The online version of this article includes the following figure supplement(s) for figure 3:

**Figure supplement 1.** Correlation statistics (Rho) for gene evolutionary rate with PC1 3L phenotype.

between known cancer genes from *Bailey et al. (2018)* and found that tumor suppressor genes had significantly lower correlation statistics (rho) than all genes (Wilcoxon rank-sum p-value=4.114e-8) while oncogenes had no significant difference in correlation statistics compared to all genes (Wilcoxon rank-sum p-value=0.3745) (*Figure 3—figure supplement 1*). This may indicate preferential purifying selection on tumor suppressor genes in large, long-lived species for lower cancer incidence. Considering that these findings, our 3L results can be naturally interpreted in the context of Peto's Paradox (*Peto, 2016*). The paradox reasons as follows: if all cells have a similar probability of undergoing a malignant transformation, organisms with more cells should have a greater risk of developing cancer. However, empirical cancer rates do not vary with body size (*Peto, 2016*), which implies that larger animals harbor mechanisms to suppress cancer rates. Top 3L constrained pathways are associated with multiple cancer control mechanisms, including DNA repair, cell cycle control, cell death, and immune function (*Figure 3B* and *Figure 3C*). A normal cell's transformation to malignancy involves failure of all these processes, and our analysis suggests that 3L animals are invested in the maintenance of each of their associated pathways through increased purifying selection. Based on enrichment and permulation results, we can infer that cell cycle fidelity, an early step in cancer development, is most important over evolutionary time scales for 3L species. Further, there is no evidence for enrichment of pathways associated with metastasis and angiogenesis, later steps in cancer development. This finding suggests that large, long-lived species have experienced increased selective pressure to protect pathways involved in early cancer stages but not later stages, perhaps because the most severely negative fitness impacts of cancer are felt earlier in its development. Species-specific cancer control mechanisms have been identified in individual species, such as increased *TP53* copy number in elephants (*Sulak et al., 2016*), but we show here that investment is cancer control is key to longevity across the entire mammalian phylogeny because top enriched pathways for the 3L phenotype do not depend on a handful of species (*Figure 1—figure supplement 3* and *Figure 1—figure supplement 2*).

An additional pathway that shows a strong signal of increased constraint with the 3L phenotype is the insulin-like growth factor (IGF1) signaling pathway (*Figure 4*), which deserves special consideration. Perturbations of IGF1 signaling result in changes in lifespan and body size in diverse organisms (*Johnson et al., 2013*; *Kimura et al., 1997*; *Stout et al., 2013*), which suggests that the IGF1

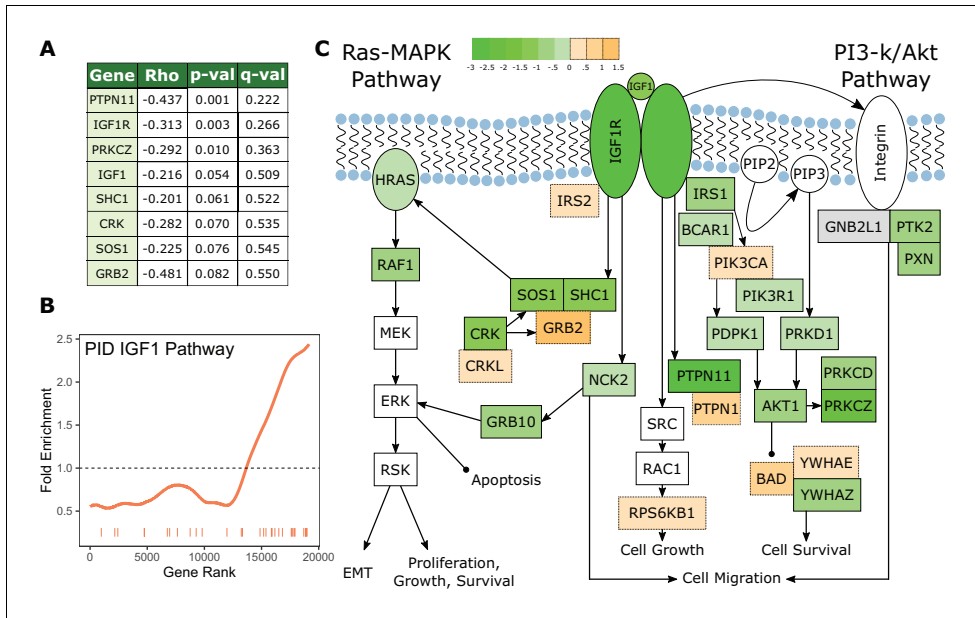

**Figure 4.** The IGF1 signaling pathway is under increased evolutionary constraint in long-lived, large-bodied mammals. (A) IGF1 signaling pathway genes are significantly correlated with change in the 3L phenotype. (B) The IGF1 signaling pathway is significantly enriched for increased evolutionary constraint in large, long-lived species. The barcode indicates ranks of genes in the pathway within the list of all pathway-annotated genes. The worm indicates enrichment as calculated by a tricube moving average, a type of moving average in which values near the end of the sliding window are down weighted to reduce the effect of extreme values in any given window. The dashed horizontal line marks the null value indicating no enrichment. (C) The IGF1 signaling pathway contains many genes whose evolutionary rates are negatively correlated with the 3L phenotype. Shading indicates the Rho-signed negative log p-value for the correlation. Genes in white are not included in the IGF1 pathway annotation used to calculate pathway enrichment statistics, but they are included in the diagram for sake of completeness. The GNB2L1 gene (gray) is in the IGF1 pathway annotation, but correlation statistics were not calculated for that gene because too few branches in the gene tree met the minimum branch length cut-off.

pathway may be a source of innovation underlying the evolution of the 3L trait. However, we find that IGF1 pathway genes in fact evolve more slowly and are thus under increased *purifying* selection in large, long-lived species. The magnitude of this signal is quite striking; IGF1 and IGF1R are ranked 1014 and 94 respectively for increased constraint and several other pathway members are near the top (see *Figure 3A*). A possible explanation is that the IGF1 pathway plays an important role in cancer control (*Larsson et al., 2005*), and this cancer-related signal of constraint dominates any adaptive signal related to lifespan. There are also further reasons to believe that the IGF1 pathway is not the main source of the 3L trait. Across the mammalian phylogeny, lifespan is strongly correlated with body size, but genetic perturbations in the IGF pathway result in longer-lived individuals that are of smaller size (*Holzenberger et al., 2003*; *Sutter et al., 2007*) thus decoupling the two traits. This strongly suggests that changes in the IGF1 pathway are unlikely to drive the natural evolution of the large, long-lived phenotype, which is established through a different, yet-unknown mechanism.

For the ELL phenotype, we find a smaller, more focused set of enriched genes and pathways. Of those significantly enriched constrained pathways for ELL, we see some overlap with functional groups represented in results from the 3L phenotype, notably immune-related and DNA repair pathways (*Figure 5A*). However, although the functional groups are the same, the pathways contained within them differ between the two phenotypes (File S2, File S3). In particular, the only significantly constrained DNA repair pathways for the 3L phenotype involve Fanconi's anemia, while the ELL phenotype shows significantly constrained DNA repair pathways for a variety of repair functions (*Figure 5*). Such pathways stand out not only because of the connection between DNA repair and cancer control, but also because of the observed relationship between DNA repair and aging independent of cancer incidence. This relationship can be demonstrated experimentally by creating double-stranded DNA breaks in laboratory mice to induce an aging phenotype (*White et al., 2015*).

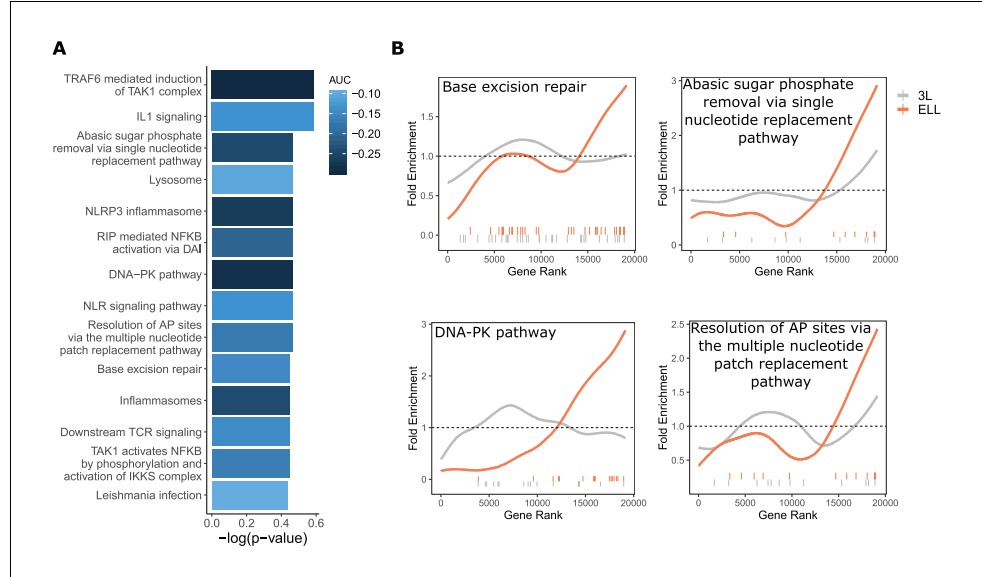

**Figure 5.** DNA repair pathways are under increased evolutionary constraint in mammals that are exceptionally long-lived given their body size. (**A**) Significantly enriched pathways under increased constraint in species with larger values of the ELL phenotype. Bar height indicates the negative log permulation p-value for each pathway, and the color of bars indicates the pathway enrichment statistic. (**B**) DNA repair pathways are more significantly enriched for increased evolutionary constraint in species with large values of the ELL phenotype than species with large values of the 3L phenotype. The barcodes indicate ranks of genes in the pathways within the list of all pathway-annotated genes. The worms indicate enrichment as calculated by a tricube moving average, a type of moving average in which values near the end of the sliding window are down weighted to reduce the effect of extreme values in any given window. The dashed horizontal lines mark the null value indicating no enrichment.

There is also evidence that DNA damage causes dysregulation of the cellular chromatin state and thus can contribute to aging even in post-mitotic cells (*Oberdoerffer et al., 2008*; *Shi and Oberdoerffer, 2012*).

In addition to DNA repair-related pathways, we also noted pathways related to NFKB signaling for which overexpression in downstream targets has been associated with aging. Experimental evidence suggests a connection between NFKB signaling and DNA repair through sirtuins, a chromatin regulator family that has already been implicated in lifespan control (*Howitz et al., 2003*; *Mao et al., 2012*). Sirtuins mediate DNA damage-induced dysregulation and are also responsible for silencing NFKB-regulated genes, thus connecting the two processes (*Salminen et al., 2008*). Overall, our analysis strongly suggests that fidelity in DNA repair and NFKB signaling contributes to the fitness of ELL species, indicating that these pathways may be a fruitful avenue for aging research and intervention.

## Discussion

We employed an evolutionary rates-based method as an unbiased, genome-wide, pan-mammalian scan to identify genes and pathways that evolve significantly slower in long-lived large-bodied species (3L) and species that are exceptionally long-lived given their body size (ELL). Pathways related to cancer control, including cell cycle, DNA repair, cell death, and immunity, evolve significantly slower in 3L species, which suggests that cancer resistance is an important functionality the enable to evolution of large and long-lived species. Alternatively, a broader set of DNA repair genes and a more focused set of immune genes related to NFKB signaling evolve significantly slower in ELL species, both of which may be linked to effective DNA repair in order to preserve chromatin state.

Our analysis differs from previous efforts in both methodology and results. Firstly, we do not consider individual amino acid changes within a protein as the unit of convergence, but rather we calculate the overall evolutionary rate of a protein on each branch of a phylogeny and consider these

evolutionary rates as the unit of convergence. Secondly, unlike previous studies that focused on lineage- or species-specific changes, we look for correlations between evolutionary rates and quantitative life history traits across the entire phylogeny. This pan-mammalian approach allows us to generalize our findings to describe evolutionary trends throughout all mammals. We also draw a careful distinction between absolute lifespan and relative lifespan, which allows us to make district conclusions about the 3L and the ELL phenotypes.

The most important distinction between our work and previous work is that prior studies began with the assumption that some genes must be under positive selection in association with evolution of extended lifespan. From our unbiased analysis, we find that in fact the strongest signal of evolutionary convergence is that of *increased* constraint on certain genes and gene families in long-lived species (see *Figure 3—figure supplement 1*). While some of the pathways have been reported previously (such as cell cycle control, DNA repair, telomerase repair, and IGF1 signaling) our finding is actually the *opposite* of (though not necessarily contradictory to) the positive selection signal that has been emphasized in previous work, which from the perspective of evolutionary rate is decreased constraint. In fact, we find comparatively few genes and no significant pathway enrichments for the opposite trend – faster relative rates in longer lived species – which would correspond to genes potentially under positive selection in longer-lived species. We therefore focus our discussion on genes and pathways evolving slower in species with large values for the 3L and ELL traits. We considered these genetic elements to be important to the evolution of the 3L and ELL traits because they have been protected from accumulating deleterious mutations and therefore evolve slower.

Despite considerable previous work on genes under positive selection in species with extreme lifespan traits, our analysis did not recover any positively selected pathways in the pan-mammalian analysis. However, for any new trait to arise, some corresponding genetic changes must occur. It is possible that many of the molecular innovations that lead to increased lifespan are species-specific and thus would not be detected by our method. However, it is also true that our relative evolutionary rates method is much more suited to detecting the kind of long-term convergent increases in purifying selection that we observe. *Figure 2* represents a simple schematic history of the evolution of a trait and corresponding rate changes. During the establishment of a new trait, some genes experience a brief period of positive selection which generates the molecular innovation to create the new trait, after which the derived sequence will again be subject to purifying selection. Using extant species as a genomic reference for evolutionary history, we can only infer the average rate over the entire history and the brief period of positive selection results in a small, potentially undetectable signal. On the other hand, we can hypothesize another class of genes which become more important as a trait is established and thus experience continuously increased purifying selection after the trait exists. These are the trait 'enabling' as opposed to the trait 'establishing' genes. Because the rate change in the case of trait-enabling genes is permanent, it has a greater impact on the inferred average rate observable from extant data. Our analyses strongly support that in the case of extended longevity, such 'enabling' genes not only exist but are also convergent across independent trait change events.

We have interpreted the rate correlation from the prospective of species with large phenotype-values, thereby assuming phenotype changes and rate changes coincide. The implicit choice of default rate does not affect the final gene ranking on which our analysis is based, but it does have one important consequence. While decreases in rate are easily interpretable as increased constraint, rate increase can be due to relaxation of constraint or positive selection. We address this asymmetry simply by testing both possibilities. We test the relative accelerated genes for positive selection in the long-lived species and the relatively decelerated genes for positive selection in the short-lived species. We find no evidence of systematic positive selection in either direction.

We interpret our strongest pathway enrichment signals as indicating that cancer control is important for enabling evolution of the 3L phenotype, which is in agreement with previous work that has found similar changes in expression levels of cell cycle and immune function genes in both cancerous and aging cells (*Chatsirisupachai et al., 2019*). However, there may be alternative explanations. Specifically, differential pathogen pressure is expected to affect some of the same pathways. The connection to immune pathways is clear, and viruses are known to co-opt the cell cycle. TP53 and cell death are likewise important mechanisms by which cells control viral infection. All of these pathways may thus be under increased constraint in species more likely to experience viral infections. However, we believe these explanations are unlikely for several reasons. Firstly, we do not see an

enrichment of genes that are specialized for virus control, such as MX1, OAS1, DHX58, and genes activated by type I interferon. Secondly, the interaction between pathogen pressure and evolutionary rates has been studied extensively and it is found that it typically drives adaptive changes in pathogen interacting protein as pathogens are often unique to a species (*Kosiol et al., 2008*; *Sackton et al., 2007*; *Shultz and Sackton, 2019*). This is at odds with the increased constraint of immune genes that we observe. Finally, the probability of cancer increases with age and there is theoretical and empirical evidence that importance of cancer resistance increases with body-size and lifespan. On the other hand, for humans the probability of dying from many infections decreases sharply in early life (though increases again at post-reproductive ages) (*Palmer et al., 2018*) suggesting that infection resistance should not be preferentially selected for in long-lived species.

Overall, the genes and pathways we identified whose constraint is negatively correlated with the 3L and ELL traits encompass functionalities important for the evolution of extended lifespan, and they therefore represent candidate genes and pathways for further experimental exploration. Importantly, such genes were uncovered using an unbiased, genome-wide pan-mammalian scan. As such, these results point to keys to exceptional longevity that are not specific to one or a handful of species, but that are universal across mammals.

## Materials and methods

RERconverge measures relationships between relative evolutionary rates of orthologous genes and phenotype values based on a set of gene trees whose branch lengths represent protein evolutionary rates and a set of phenotype values, either binary or continuous. The RERconverge package is freely-available for use at https://github.com/nclark-lab/RERconverge along with walkthroughs for beginner users (*Kowalczyk et al., 2018*). The original RERconverge method was successfully used to find genes undergoing convergent evolutionary rate shifts in marine mammals (*Chikina et al., 2016*) and subterranean mammals (*Partha et al., 2017*), and recent statistical improvements have made RERconverge more robust and have given it more power to detect such rate shifts (*Partha et al., 2018*). Here, the RERconverge methods were extended to use on continuous traits.

To perform the RERconverge analysis, we first used PAML (*Yang, 2007*) to create maximum likelihood gene trees whose branch lengths represent evolutionary rates by means of number of amino acid substitutions. Gene trees were generated for 19,149 amino acid alignments of orthologous genes from the 62 mammal species from the UCSC 100-way alignment (*Blanchette et al., 2004*; *Harris, 2007*; *Kent et al., 2002*) using the hg19 genome, all of which are available at http://genome.ucsc.edu/. Trees were read into R using the readTrees() function from RERconverge. This step also estimated a master tree with branch lengths that represented the average branch lengths across many gene trees. For all further steps, only 61 species were used as listed in *Supplementary file 1*; the cape golden mole was excluded because longevity data were not available for it. Relative evolutionary rates corrected for genome-wide evolutionary rates for each species as well as corrected for branch length heteroskedasticity using a weighted regression approach (*Partha et al., 2018*) were calculated using getAllResiduals() with weight = T, scale = T, cutoff = 0.001, and useSpecies = species names for the 61 mammals species.

Relative evolutionary rates (RERs) quantify the deviation in evolutionary rate of a protein along a specific phylogenetic branch from proteome-wide expectations. Negative RERs indicate fewer substitutions than expected due to increased constraint. Positive RERs correspond to more substitutions than expected, which could arise due to relaxation of constraint or positive selection. To correlate RERs with lifespan phenotypes, we use phenotypic change along phylogenetic branches computed from maximum likelihood ancestral state reconstruction (*Revell, 2012*). This transformation is equivalent to phylogenetically independent contrasts (*Felsenstein, 1985*) and thus removes phylogenetic dependence from the phenotype values.

We used the char2Paths() function on each phenotype with trees read in using readTrees() to create a phenotype vector that represented the predicted difference in phenotype values between each species and its ancestor. The char2Paths() function first uses maximum likelihood estimation through the fastAnc() function from phytools (*Revell, 2012*) and then subtracts the values between connected pairs of species to calculate a phylogenetically-independent measure of the change in the phenotype over evolutionary time. Finally, the getAllCor() function was run with each phenotype and the RER matrix with method = 'p', min.pos = 0, and winsorizeRER = 3 to perform Pearson

correlations with RERs winsorized such that the two most extreme RER values are set to the value of the third most extreme RER. See *Figure 1—figure supplement 4* for evolutionary rate-phenotype scatterplots with high correlation values.

The gene lists produced from the RERconverge correlation analyses were used to calculate pathway enrichments for all canonical pathways from mSigDB (*Liberzon et al., 2011*) and Mouse Genome Informatics (MGI) functional annotations (*Blake et al., 2003*; *Eppig et al., 2015*). For each pathway, a Wilcoxon Rank-Sum statistic was calculated to compare the sign of Rho times the negative log p-value for correlations of genes in the pathway to the same measure for all genes included in a pathway annotation using getStat() from RERconverge to calculate the sign of Rho times the negative log p-value for correlations for each gene and fastwilcoxGMTall() to quickly calculate an approximation of the Wilcoxon Rank-Sum statistic.

## Permulation analysis (Phylogenetically-Restricted Permutations)

In addition to calculating theoretical enrichment statistics, we developed a novel phylogenetically-restricted permutation strategy dubbed 'permulations' to calculate empirical p-values for each pathway enrichment statistic. Permulations are a combination of permutations and simulations in reference to the strategy for generating null phenotype values to use to generate null statistics. To perform permulations, first phenotype values (3L and ELL) were simulated for each species using phylogenetic simulations, and then original phenotype values were reassigned to the species based on the rank of the simulated values (see *Figure 1—figure supplement 5*). Simulations were performed using the geiger library in R using a Brownian motion approach and the average evolutionary rate tree created by RERconverge (*Harmon et al., 2008*). After creating new phenotype values, RERconverge analyses were performed to calculate correlations between evolutionary rates of genes and phenotype values, and enrichment statistics were calculated using these gene results. The process of creating new phenotype values and calculating enrichment statistics was repeated 1000 times for each phenotype and the empirical p-value for pathway enrichment was calculated as the proportion of times the permulation enrichment statistic was greater than the enrichment statistic calculated using real phenotype values.

Permulations are an attractive option to calculate empirical p-values because they use fabricated phenotypes that are independent of RERs but respect the underlying phylogenetic relationships between species (due to use of phylogenetic simulations) while maintaining the original range of the data (due to rank-based assignment of true phenotype values). These types of analyses are a common tool in genomics because they allow for control of the dependence across tests caused by inter-correlations among genes. Permulations are highly analogous to permutations commonly performed in differential expression analysis, and we use them for the same purpose, namely to calculate empirical pathway-level statistics (*Subramanian et al., 2005*). This step is critical for evaluating pathway enrichment results because many pathways have elevated RER correlations even when conditioned on the phenotype of interest. A pathway with high RER correlation among its genes is more likely to show up as enriched when using a test that assumes gene independence, such as the Wilcoxon Rank-Sum test. Permulations allow us to generate an empirical null distribution for pathway enrichment statistics to correct for interdependence among gene ranks. Indeed, we find multiple pathways that show significant enrichment using a Wilcoxon Rank-Sum test but insignificant empirical p-values using permulations. We also show that permulation p-values, which take phylogeny into account, are more conservative than permutation p-values, which ignore phylogenetic dependence, and equally as conservative as the phylogenetic simulation p-values (see *Figure 1—figure supplement 6*).

## Positive selection tests

In genes significantly correlated with the 3L and the ELL phenotypes, we investigated evidence for relaxation of constraint and positive selection on trait-defining foreground branches using phylogenetic models of codon evolution. We did so using a representative subset of the full mammalian phylogeny (see *Figure 1—figure supplement 7*). Trait-defining foreground branches were specified independently for the two phenotypes along both axes of trait values – positive and negative. In total we have four sets of trait-defining foreground branches, namely positive 3L, negative 3L, positive ELL, and negative ELL. *Figure 1—figure supplement 8* shows the phylogeny of species used for this analysis and the four sets of trait-defining foreground branches. Species were selected for

the representative subset based on three criteria: 1) all species were placental mammals (the monotreme and marsupials were excluded), 2) species with highest and lowest 3L and ELL phenotype values were included and used as foreground species in their respective tests, and 3) closely-related non-foreground species were included. Only placental mammals were used because the inference of positive selection can be confounded in the non-placental clade due to the long divergence time between the clades and hence saturation of synonymous sites for many genes. Foreground species were selected based on 3L and ELL values. A subset of non-foreground species was selected to correct for over- and under-sampling in particular clades while maintaining valid 'outgroups' for foreground species. This subset also eliminated species that were essentially duplicates in terms of tests for positive selection. For example, the primate clade is sampled much more than other clades, that is more primate genomes are available. Additionally, because most of these species are non-foreground species, their presence has little impact on positive selection tests. To correct for the oversampling, we used five out of twelve primates in our subtree, three of which were foreground species (human, marmoset, and squirrel monkey) and two of which were outgroups (rhesus as an outgroup to human and bushbaby as an outgroup to the whole clade). We used similar logic to select outgroup species throughout the rest of the phylogeny.

We inferred the significance of relaxation of constraint on each foreground branch set using likelihood ratio tests (LRT) between Branch-site Neutral (BS Neutral) and its nested null model M1 (sites neutral model) in PAML (*Yang, 2007*). Similarly, we performed LRTs between branch-site selection model (BS Alt Mod) and its null BS Neutral were used to infer positive selection on the foreground branches. Probabilities for each of these two LRTs were estimated using the chi-square distribution with 1 degree of freedom. We additionally inferred significance of mammal-wide relaxation of constraint and positive selection using the LRTs between M8A (Neutral model) vs M0 (null model) and M8 (positive selection model) vs M8A respectively. Prior to performing the mammal-wide tests in genes corresponding to each of the four foreground branch sets, we removed the corresponding foreground branches, allowing us to obtain unbiased estimates for significance of relaxation of constraint and positive selection from only the background mammalian branches.

## Species robustness through subtree analysis

We were interested in assessing the sensitivity of our results to the choice of mammalian species used in our analyses. The mammalian genomes available represent a subset of not only all extant mammals, but all mammal species that have ever existed. Therefore, the genomes used in this analysis represent an incomplete, and perhaps even a biased representation of mammal species. Since we would like to extend our conclusions to pertain to mammals in general, we sought to quantify the effects of our incomplete data on our gene and pathway results.

To do this, we created subsets of our data that contained fewer species. These subsets had 10, 20, 30, 40, 50, 60, 70, and 80 percent of species randomly removed (6, 12, 18, 24, 30, 36, 42, and 48 species removed out of 61 total, respectively) with ten random subsets created for each species removal level. We then ran the standard RERconverge analysis on these subsets to acquire correlation statistics representing the relationship between evolutionary rate of each gene across species and longevity phenotypes in those species. We also ran enrichment analyses on the subset gene results to acquire pathway enrichment statistics. After performing these analyses, we calculated correlations between results from our full dataset and results from the subsets, where results were quantified as the negative log p-value times the sign of the statistic.

We further tested the sensitivity of our results to specific species presence/absence by performing targeted species removal of species groups with which we expected to have other phenotypes confounded with body size, namely marine mammals, and species that may have non-convergent genetic mechanisms for lifespan extension, namely bats and the naked mole rat. We created two new data subsets that only contained non-marine species (dolphin, manatee, killer whale, walrus, and Weddell seal were removed) and only non-bat and non-naked-mole-rat species (megabat, black flying-fox, microbat, David's myotis bat, big brown bat, and naked mole-rat were removed). We then performed the standard RERconverge analysis to find correlations between the evolutionary rates of genes across species and longevity phenotypes in those species. We also performed pathway enrichment analyses on the gene results.

We found good correlation of both enrichment and correlation statistics between subset results and full dataset results based on negative log p-values times the sign of the statistic (*Figure 1—*

*figure supplement 3*). For data subsets in which a proportion of species were randomly eliminated, there was upwards of a median 60% correlation between results even with half all species removed from our analyses, which indicates that randomly removing species did not significantly change our results. These findings are quantified numerically as an among-subtree variance in *Supplementary file 4* and *Supplementary file 5*. We also compared full dataset results to results from targeted data subsets without marine mammals and without bats and the naked mole-rat by quantifying correlations between their negative log p-values times the sign of the statistic for gene correlations and pathway enrichment statistics (*Figure 1—figure supplement 8*). From these comparisons, we found a strong relationship between full dataset results and targeted subset results, which indicates that presence or absence of potentially problematic species such as marine mammals, bats, and the naked mole-rat is not strongly impacting our results. Quantitative representations of these findings are available in *Supplementary file 4* and *Supplementary file 5* Together, these findings indicate that our results are not species, clade, or species subgroup-specific, but instead represent trends across all mammals.

## Alternate tree topology analysis

In addition to determining whether individual or groups of species were disproportionately driving our conclusions, we were also interested in verifying that the fixed tree topology used for analyses was not affecting results. To do so, we reran analyses using alternate plausible topologies representing different ancestral relationships among species (trees shown in *Figure 1—figure supplement 9*). Differences in the alternate tree topologies represent points of potential incomplete lineage sorting that lead to uncertainty, they and are thus reasonable alternatives compared to the original tree used for analyses.

RERconverge was run using gene trees generated using the alternate topologies and results were compared to results using the original tree. As shown in *Figure 1—figure supplement 10*, there is a strong correlation between all sets of results, which indicates that uncertainties in tree topology do not strongly affect results. This is true for very similar topologies (Robinson-Foulds distance 6) and fairly different topologies (Robinson-Foulds distance 22).

## Phylogenetic trees

Full Tree (61 mammals):

(((((((((((((Human:0.005957477577,Chimp:0.006721826689):0.001382639829, Gorilla:0.007765177171):0.005572327638,Orangutan:0.0164503644):0.002187630666,Gibbon:0.01770384793):0.007043113559,(Green_monkey:0.007693724903,((Crab-eating_macaque:0.001292320552,Rhesus:0.00713015786):0.002951690224, Baboon:0.005199240711):0.002049749893):0.01566263562):0.0135408115,(Marmoset:0.02474184521,Squirrel_monkey:0.02096868307):0.02784675729):0.04299750653,Bushbaby:0.108738222):0.01379370868,((((((Guinea_pig:0.09048639907,(Chinchilla:0.05332953299,Brushtailed_rat:0.08476954109):0.01287861561):0.02118937782,Naked_mole-rat:0.08588673524):0.07432515556,Squirrel:0.08896424642):0.006291577528,((((Chinese_hamster:0.04084640027,Golden_hamster:0.04456203524):0.02314125062,Prairie_vole:0.06932402649):0.01947113467,(Mouse:0.05273642272, Rat:0.05576007402):0.04435347588):0.08380065137,Lesser_Egyptian_jerboa:0.1438649666):0.04270536633):0.01663675397,(Pika:0.1256544445,Rabbit:0.07131655591):0.06535533418):0.009050428462,Chinese_tree_shrew:0.1191189141):0.003894252213):0.01425600689,(((((Panda:0.03854019703,((Weddell_seal:0.02002160645,Pacific_walrus:0.02064385875):0.01734764946,Ferret:0.04613997497):0.002879093616):0.009005888384,Dog:0.05339127565):0.01185166857, Cat:0.05020331605):0.03285617057,((((((Cow:0.02168740723,((Domestic_goat:0.01157093136, Sheep:0.01246322594):0.0049716126,Tibetan_antelope:0.01522587482):0.01465511149):0.0662523666,(Killer_whale:0.006371664911,Dolphin:0.01086552617):0.06014682602):0.01216198069,Pig:0.0796745271):0.006785823323,(Bactrian_camel:0.01240650215,Alpaca:0.01096629635):0.06374554586):0.02551888691,(White_rhinoceros:0.04977357056,Horse:0.061454379):0.02510111297):0.00331214686,((Big_brown_bat:0.03248546656,(Davids_Myotis_bat:0.02344332842,

Microbat:0.01567729315):0.02193849809):0.09455328094,(Black_flying-fox:0.005833353548,Megabat:0.01611220178):0.07567400302):0.02385546003):0.002057771224):0.004845253848,(Star-nosed_mole:0.1239823369,(Hedgehog:0.1696142244,
Shrew:0.1934205791):0.02079474546):0.0235875333):0.01477733374):0.01316436193,(((((Cape_golden_mole:0.1017903453,Tenrec:0.1749615473):0.01592632003,Cape_elephant_shrew:0.1516860647):0.006610995228,Aardvark:0.08326528894):0.008243787904,(Elephant:0.06812658238,Manatee:0.06198982615):0.0224994529):0.03384011363,Armadillo:0.1342602666):0.005989703247):0.2206952867,((Wallaby:0.1270943532,Tasmanian_devil:0.09944141622):0.02717055443,Opossum:0.1181200712):0.1802966572):0,
Platypus:0.4322118716);

Branch Site Tree (34 placental mammals):
((((((Human:0.02214318927,Rhesus:0.0277942336):0.0135408115,(Marmoset:0.02474184521,
Squirrel_monkey:0.02096868307):0.02784675729):0.04299750653,Bushbaby:0.108738222):0.01379370868,(((Chinchilla:0.08739752642,Naked_mole-rat:0.08588673524):0.07432515556,Squirrel:0.08896424642):0.006291577528,((((Chinese_hamster:0.04084640027,Golden_hamster:0.04456203524):0.02314125062,Prairie_vole:0.06932402649):0.01947113467,(Mouse:0.05273642272,
Rat:0.05576007402):0.04435347588):0.08380065137,Lesser_Egyptian_jerboa:0.1438649666):0.04270536633):0.02958143464):0.01425600689,((((Weddell_seal:0.02002160645,Pacific_walrus:0.02064385875):0.04108430003,
Cat:0.05020331605):0.03285617057,(((((Cow:0.02168740723,(Sheep:0.01743483854,Tibetan_antelope:0.01522587482):0.01465511149):0.0662523666,Killer_whale:0.06651849094):0.01216198069,
Pig:0.0796745271):0.03230471024,Horse:0.08655549197):0.00331214686,((Big_brown_bat:0.03248546656,(Davids_Myotis_bat:0.02344332842,Microbat:0.01567729315):0.02193849809):0.09455328094,Black_flying-fox:0.08150735657):0.02385546003):0.002057771224):0.004845253848,(Star-nosed_mole:0.1239823369,(Hedgehog:0.1696142244,
Shrew:0.1934205791):0.02079474546):0.0235875333):0.01477733374):0.01316436193,((Cape_elephant_shrew:0.1665408478,(Elephant:0.06812658238,Manatee:0.06198982615):0.0224994529):0.03384011363,Armadillo:0.1342602666):0.005989703247);

## Data and materials availability

All data and code are publicly available, provided in the main text or supplementary materials, or available through the RERconverge package on GitHub at https://github.com/nclark-lab/RERconverge. (*Kowalczyk et al., 2019*; copy archived at https://github.com/elifesciences-publications/RERconverge).

## Acknowledgements

We would like to thank Dr. Andreas Pfenning and Dr. Dennis Kostka for helpful discussion and feedback, as well as all members of the Clark and Chikina labs.

## Additional information

### Funding

| Funder | Grant reference number | Author |
| --- | --- | --- |
| National Institutes of Health | R01HG009299 | Nathan L Clark<br>Maria Chikina |
| National Institutes of Health | U54 HG008540 | Nathan L Clark<br>Maria Chikina |
| Howard Hughes Medical Institute | T32 EB009403 | Amanda Kowalczyk |

The funders had no role in study design, data collection and interpretation, or the decision to submit the work for publication.

## Author contributions
Amanda Kowalczyk, Conceptualization, Data curation, Software, Formal analysis, Investigation, Visualization, Methodology; Raghavendran Partha, Conceptualization, Resources, Data curation, Software, Formal analysis, Supervision, Funding acquisition, Visualization, Methodology, Project administration; Nathan L Clark, Conceptualization, Resources, Supervision, Funding acquisition, Project administration; Maria Chikina, Conceptualization, Resources, Data curation, Software, Supervision, Funding acquisition, Visualization, Methodology, Project administration

## Author ORCIDs
Amanda Kowalczyk (iD) https://orcid.org/0000-0002-9061-1336
Raghavendran Partha (iD) http://orcid.org/0000-0002-7900-4375
Nathan L Clark (iD) http://orcid.org/0000-0003-0006-8374

## Decision letter and Author response
Decision letter https://doi.org/10.7554/eLife.51089.sa1
Author response https://doi.org/10.7554/eLife.51089.sa2

# Additional files
## Supplementary files
• Supplementary file 1. Phenotype values. Includes longevity, body size, 3L, and ELL values for the 61 mammal species used for analyses.

• Supplementary file 2. Pathway enrichment analysis results. Pathway enrichment results for both the 3L and ELL phenotypes using MSigDB canonical pathway annotations.

• Supplementary file 3. Pathway clustering for 3L enriched pathways. Pathway group membership for the 3L phenotype as depicted in *Figure 2*.

• Supplementary file 4. Gene-phenotype evolutionary rate correlation and subtree analysis gene-phenotype correlation results. Correlation statistics between gene evolutionary rates and phenotype values as obtained from RERconverge for both the 3L and ELL phenotypes separated by correlation direction. Also included are statistics from species removal analyses.

• Supplementary file 5. Subtree analysis pathway enrichment results. Pathway enrichment statistics from species removal analyses for both the 3L and ELL phenotypes using MSigDB canonical pathway annotations.

• Supplementary file 6. Branch-site model tests for positive selection results. Results from branch-site model tests on genes to detect genes under positive selection in species at both extremes of the 3L and ELL phenotypes.

• Transparent reporting form

## Data availability
Genome alignments are publicly available through the UCSC genome browser - processed alignments in the form of phylogenetic trees that were used for this study are available via the RERconverge package on GitHub. Code for the RERconverge package and vignettes instructing proper use are publicly available on GitHub (https://github.com/nclark-lab/RERconverge; copy archived at https://github.com/elifesciences-publications/RERconverge). Longevity phenotype data is available as supplementary material.

The following previously published dataset was used:

| Author(s) | Year | Dataset title | Dataset URL | Database and Identifier |
|-----------|------|---------------|-------------|-------------------------|
| Blanchette M, Kent WJ, Riemer C, Elnitski L, Smit AFA, Roskin KM, Baertsch R, Rosen- | 2003 | UCSC 100-way alignment | http://hgdownload.cse.ucsc.edu/goldenPath/hg19/multiz100way/alignments/knownCanonical.exonAA.fa.gz | UCSC genome browser, knownCanonical.exonAA |

bloom K, Clawson H, Green ED, Haussler D, Miller W

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
