## [Decision Letter]

**Acceptance summary:**

This study provides important insights into the evolution of lifespan across a range of mammalian species and identifies several genes under increased evolutionary constraint in long-lived species.

**Decision letter after peer review:**

Thank you for submitting your article "Pan-mammalian analysis of molecular constraints underlying extended lifespan" for consideration by *eLife*. Your article has been reviewed by two peer reviewers, one of whom is a member of our Board of Reviewing Editors, and the evaluation has been overseen by Diethard Tautz as the Senior Editor. The reviewers have opted to remain anonymous.

The reviewers have discussed the reviews with one another and the Reviewing Editor has drafted this decision to help you prepare a revised submission.

Summary:

Kowalczyk and colleagues report on a comparative genomics approach to understand potential mechanisms of lifespan determination in mammals. They consider protein evolutionary rates across a wide range of mammalian species and identify signatures suggesting that genes related to cell cycle, DNA repair, cell death, the IGF1 pathway, and immunity are under evolutionary constraint in long-lived mammalian species. The unbiased approach is a particular strength of this study, as much of the prior work in this area has involved targeted studies aimed at querying specific mechanisms/pathways in long-lived species. Although the conclusions are a bit short on new insight into potential conserved mechanisms of longevity determination, they generally make biological sense and add further support for the necessity of robust anti-cancer mechanisms to evolve in parallel with long lifespans in large mammals.

Essential revisions:

1) Parts of the selection analysis should be expanded to consider shorter lifespans. A first assumption seems to be that longer lifespans tend to be achieved through positive selection; while shorter lifespans tend to be achieved through relaxed purifying selection. This fits with Mutation Accumulation and Antagonistic Pleiotropy theories in the sense that shorter lifespans would evolve due to the relative lack of efficient purifying selection at older ages; while one would need to fix advantageous mutations to lengthen lifespan. This seems correct in general, even if, to the best of our knowledge, no one has shown this to always be the case. A second assumption seems to be that once long lifespans have been achieved, purifying selection will keep them this way and, hence, they will come with would come with constrained rates of protein evolution in similar pathways. That is what they set out to detect, since it would be indicative of convergent evolution at the molecular level.

However, there is the additional possibility that once a species has achieved a shorter lifespan (for instance, due to a trade-off to increase early-age reproduction) purifying selection keeps it this way. It may be the case, thus, that increased constraint would also be detected in similar pathways in species with lower 3L and ELL values. We think this study is particularly suited to address this point with some rigor. So please, look at "more constrained in short lived" pathways and tell us what you see.

2) The multiple testing correction needs a better explanation. Reading the Materials and methods, it would seem that "permulations" only apply to pathways, "…to calculate empirical values for each pathway enrichment statistic", only pathway enrichment is mentioned there). However, it seems (from the first paragraph in the Results section and from Figures 1C and 1D) that the first thing done with "permulations" is to see if they have an excess of significant p-values in the set of individual gene correlations. Is that so? We think clarifying this is critical since, of course, any distribution of so many correlations between RERs and phenotypes will have extremes and significant p-values. Incidentally, we would suggest including conventional QQ-plots, at least as supplementary information.

3) Related to the previous point, we are fine with using RERs, 3L and ELL. However, we are not fully convinced that "permulations" are so conservative. First, using the "average evolutionary rate tree" is likely to generate mismatches with closely related species where incomplete lineage sorting or gene flow have been relevant phenomena. In these cases, the distribution of observed correlation p-values may be different from the simulated (permulated) distributions. Second, we know that rates of evolution (of both, molecules and phenotypes) do not follow an exact clock.

---

## [Author Response]

Essential revisions:1) Parts of the selection analysis should be expanded to consider shorter lifespans. A first assumption seems to be that longer lifespans tend to be achieved through positive selection; while shorter lifespans tend to be achieved through relaxed purifying selection. This fits with Mutation Accumulation and Antagonistic Pleiotropy theories in the sense that shorter lifespans would evolve due to the relative lack of efficient purifying selection at older ages; while one would need to fix advantageous mutations to lengthen lifespan. This seems correct in general, even if, to the best of our knowledge, no one has shown this to always be the case. A second assumption seems to be that once long lifespans have been achieved, purifying selection will keep them this way and, hence, they will come with would come with constrained rates of protein evolution in similar pathways. That is what they set out to detect, since it would be indicative of convergent evolution at the molecular level.However, there is the additional possibility that once a species has achieved a shorter lifespan (for instance, due to a trade-off to increase early-age reproduction) purifying selection keeps it this way. It may be the case, thus, that increased constraint would also be detected in similar pathways in species with lower 3L and ELL values. We think this study is particularly suited to address this point with some rigor. So please, look at "more constrained in short lived" pathways and tell us what you see.

We appreciate the reviewers’ thoroughness and extensive knowledge of aging-related theories, and we have spent considerable time discussing these ideas. One key finding in this study was that levels of purifying selection, or conservation, across species with a range of 3L and ELL phenotypes represent relative gene importance. Our broad hypothesis based on findings in this work is that latent positive selection at some point in evolutionary history drove establishment of the 3L and ELL phenotypes, while selective pressure shifts associated with trait establishment caused sustained increased purifying selection of genes supporting the newly-established trait. Since RERconverge detects average rate changes over evolutionary time, we are unable to detect periods of brief positive selection, while we are much more readily able to detect sustained purifying selection. A visual of this hypothesis is shown in Figure 1—figure supplement 1. Results from both RERconverge and branch-site models to test for positive selection were unable to detect instances of positive selection support this model. It is also possible that instances of positive selection were non-convergent, in which case they would neither be detectable to RERconverge nor relevant to generalizable conclusions about the origin of extended lifespan in mammals. We performed branch-site model tests for positive selection using species with both large and small values of both 3L and ELL phenotypes as foreground species to test genes accelerated in species with both large and small values of both 3L and ELL phenotypes, thus accounting for both correlation directions. None of the branch-site model tests detected meaningful positive selection in any foreground species, and broadly, RERconverge detected more significant negative correlations (genes evolving faster in small, short-lived species and slower in large, long-lived species) than positive correlations (genes evolving slower in small, short-lived species and faster in large, long-lived species), with no significantly enriched pathways evolving faster in species with large values of the 3L and ELL phenotypes. Therefore, although it is theoretically possible that genes could be under increased purifying selection to increase phenotypes related to evolution of small size and short lifespan, we do not observe such genes in our analysis.

Further, since long lifespan and large body size are inferred to be derived traits (the ancestral mammal is believed to have been small and short lived), evolutionary states of species with large values of 3L and ELL are likely representative of changes related to evolution of those traits, while evolutionary states of species with small values of 3L and ELL are likely representative of maintenance of ancestral states. This may explain why we do not observe these positive correlations in our analyses.

Mutation Accumulation and Antagonistic Pleiotropy theories broadly apply to lifespan on the scale of an individual rather than changes in lifespan over evolutionary time. These theories state, respectively, that accumulation of mutations throughout an individual’s lifetime eventually leads to breakdown of essential functions that drive senescence and eventual age-related death, and that functions that benefit early lifetime survival and fitness are actually detrimental in old age. We do not believe that potential reduced mutation rates in old age in large and long-lived mammals are due to effective increased purifying selection in these species, but instead that increased purifying selection on targeted genes, such as those for DNA repair and cell cycle control, facilitates improved functionalities of these genes throughout an individual’s lifespan and thus increased conservation indirectly contributes to existing hypotheses regarding aging.

We have updated the Results section as follows to further clarify that we addressed both correlation directionalities and considered both possible directions of positive selection:

“Our analysis investigates both positive and negative correlations between evolutionary rates of genes and changes in lifespan phenotypes. […] Both the gene-level and pathway-level results were highly robust to species removal (see Materials and methods, Figure 1—figure supplement 6, Supplementary file 4, and Supplementary file 5).”

We have also added Figure 1—figure supplement 1 to clarify why we detect more genetic signal for pathways under increased purifying selection than positive selection.

2) The multiple testing correction needs a better explanation. Reading the Materials and methods, it would seem that "permulations" only apply to pathways, "…to calculate empirical values for each pathway enrichment statistic", only pathway enrichment is mentioned there). However, it seems (from the first paragraph in the Results section and from Figures 1C and 1D) that the first thing done with "permulations" is to see if they have an excess of significant p-values in the set of individual gene correlations. Is that so? We think clarifying this is critical since, of course, any distribution of so many correlations between RERs and phenotypes will have extremes and significant p-values. Incidentally, we would suggest including conventional QQ-plots, at least as supplementary information.

We agree with reviewers that multiple hypothesis testing correction is a key part of our analyses since performing nearly 20,000 tests for genes will lead to false positives. Reviewers are also correct that permulation p-values were only calculated for pathways, although null permulation p-values were generated both for genes and pathways, as they are necessary in order to calculate permulation p-values (permulation p-values are the proportion of null permulation statistics more extreme than an observed statistic for a particular test, be it gene-level or pathway level). At the gene level, we perform classic multiple hypothesis testing correcting using the Benjamini-Hochberg correction. We compare this multiple hypothesis testing correction, which uses a standard uniform distribution to represent the assumed proportion of false positive p-values, to multiple hypothesis testing correction using the true empirical null hypothesis for this dataset as calculated using permulation null p-values, or p-values generated by shuffling phenotype values with respect to the underlying phylogeny. Based on this empirical null distribution generated using null permulation p-values, we find that our theoretical adjusted p-values are actually more conservative than we would expect because the distribution of empirical null p-values is slightly non-uniform and in fact slopes downward at the left end of the p-value distribution (Figures 1D and 1E).

At the pathway level, we do calculate empirical p-values on a per-pathway basis. These p-values represent the proportion of null permulation enrichment statistics that are more extreme than the observed pathway enrichment statistics, akin to bootstrap p-values. Again, these permulation p-values are multiple hypothesis testing corrected using a standard Benjamini-Hochberg correction.

Permutation analyses are a common tool in genomics because they allow one to control for the dependence across tests. They are ubiquitously used in differential expression analysis to calculate empirical FDRs for gene-level test and empirical p-values for pathway-level test. In the simplest case we are concerned with comparing two groups with different clinical phenotypes. The null hypothesis is that the phenotype differences are not reflected in the data and thus with respect to the data the two groups are interchangeable. Consequently, permutations of group labels can be used to generate an empirical null distribution while controlling for correlations across genes (which remain unperturbed). This empirical null can then be used to calculate empirical FDRs for gene level tests. Correlation among genes also must be considered for evaluating pathway level analysis. Many tests for pathway enrichment assume that genes are independent when conditioned on the phenotype, which is clearly false. Genes that are co-expressed tend to have similar differential expression statistics. Permutations can be used to generate per-pathway null test statistic distribution by controlling for correlations among pathway genes. This distribution can then be used to calculate an empirical p-value for pathway enrichment, which will typically be larger than those derived from a parametric test that assumes independence.

Our permutations pipeline follows closely what would typically be done in a gene expression context. However, in our case we cannot simply permute the species values because even under the null hypothesis these are not independent. To extend the differential expression analogy, if some patients were profiled more than once any valid permutation would need to place those samples in the same group. In our case, the species are related through the phylogeny, so we perform phylogenetically constrained permutations that ensure that the species that are phylogenetically closer are also more likely to receive similar phenotype values. As expected, this results in more conservative p-values than the unconstrained permutations (Figure 1—figure supplement 9).

The main text has been altered as follows to further clarify the role of permulations in hypothesis testing:

“Permulations are an attractive option to calculate empirical p-values because they use fabricated phenotypes that are independent of RERs but respect the underlying phylogenetic relationships between species (due to use of phylogenetic simulations) while maintaining the original range of the data (due to rank-based assignment of true phenotype values). […] We also show that permulation p-values, which take phylogeny into account, are more conservative than permutation p-values, which ignore phylogenetic dependence, and equally as conservative as the phylogenetic simulation p-values (see Figure 1—figure supplement 9).”

We have also added Q-Q plots as a supplementary figure as requested (Figure 1—figure supplement 10).

3) Related to the previous point, we are fine with using RERs, 3L and ELL. However, we are not fully convinced that "permulations" are so conservative. First, using the "average evolutionary rate tree" is likely to generate mismatches with closely related species where incomplete lineage sorting or gene flow have been relevant phenomena. In these cases, the distribution of observed correlation p-values may be different from the simulated (permulated) distributions. Second, we know that rates of evolution (of both, molecules and phenotypes) do not follow an exact clock.

We agree with reviewers that rates of evolution do not follow an exact clock, and we fully take advantage of that in order to perform our analyses to detect evolutionary rate shifts associated with phenotypic changes. Also note that we take into account broad differences in evolutionary rates on the species level by normalizing for differences in species evolutionary rates caused by factors such as generation times by quantifying evolution as a rate relative to species average evolutionary rate (thus *relative* evolutionary rates as RERs).

We additionally fully subscribe to the notion that a fixed phylogeny is a model, and although all models are wrong, some are useful and some are better than others. We strived to create the best phylogeny possible by incorporating information from two highly-regarded studies that built phylogenies as well as our own literature review of highly-contested relationships among species. Although most of the species we considered are so distantly related that we don’t expect many completely incorrect relationships in extant species, there undoubtedly was incomplete lineage sorting on internal branches of the phylogeny. In our analyses, this circumstance would be reflected by extremely short branch lengths representing little evolutionary change that would be filtered out prior to statistical analyses, thus making the analysis robust to phylogenetic inaccuracies. We have demonstrated this robustness by performing our analyses using two different phylogenies, one with a Robinson-Fould distance of 6 and one with a distance of 22 from the phylogeny used in the original study (Figure 1—figure supplement 11). These distances represent the number of “swaps” among branches that occurred in the phylogenetic topology. As highlighted in the supplementary figure, locations of swaps are at points of uncertainty, such as the relationship between the Chinese tree shrew and primates, that likely exist because of phenomena such as incomplete lineage sorting that cause confusion among species at the molecular level. There is a strong correlation between both gene-level and pathway-level statistics between the Meredith+ topology and the alternative topologies (Figure 1—figure supplement 12), which indicates that the reported results are robust to changes in tree topology representing topological uncertainties and instances of potential incomplete lineage sorting.

We have included Figure 1—figure supplements 11 and 12 to demonstrate the tree topologies tested and the correlations between results from different tree topologies.

We have also added the following text to the manuscript to describe the alternate tree topology analyses:

“Alternate Tree Topology Analysis

In addition to determining whether individual or groups of species were disproportionately driving our conclusions, we were also interested in verifying that the fixed tree topology used for analyses was not affecting results. […] This is true for very similar topologies (Robinson-Foulds distance 6) and fairly different topologies (Robinson-Foulds distance 22).”

In regards to the relationship between permulation p-values and theoretical p-values, we expect and observe differences between the two values at the level of individual pathways. Some pathways that exhibit strong enrichment according to theoretical p-values have non-significant permulation p-values, likely because the permulation p-values account for confounders related to the structure of the data that we as statisticians do not anticipate. The beauty of permulation p-values is that they represent the true null expectation for the given dataset – under the given phylogeny and set of statistical tests, we can know what the null expectation is for the statistics because we have generated thousands of “fake” (permulation) phenotype values that should not have significant statistics (because they are literally fake phenotypes). For these “fake” phenotypes, we do not expect to observe significant correlations with genes and subsequent significant pathway enrichment except in false positive cases, which may or may not occur at a rate of about 0.05 (α). If, for a particular pathway, false positive cases happen at random more often than we would expect (greater than α=0.05 of the time), that is quantified by the permulation p-values – in that case, more null permulation p-values would be significant, more null permulation statistics would be more extreme than the observed statistic, and thus the calculated empirical permulation p-value for the pathway would be higher (less significant) because we know that the probability of getting a false positive significant p-value is higher than the typical null expectation.